# Global Burden Related to Nitrous Oxide Exposure in Medical and Recreational Settings: A Systematic Review and Individual Patient Data Meta-Analysis

**DOI:** 10.3390/jcm8040551

**Published:** 2019-04-23

**Authors:** Abderrahim Oussalah, Mélissa Julien, Julien Levy, Olivia Hajjar, Claire Franczak, Charlotte Stephan, Elodie Laugel, Marion Wandzel, Pierre Filhine-Tresarrieu, Ralph Green, Jean-Louis Guéant

**Affiliations:** 1University of Lorraine, INSERM UMR_S 1256, Nutrition, Genetics, and Environmental Risk Exposure (NGERE), Faculty of Medicine of Nancy, F-54000 Nancy, France; abderrahim.oussalah@univ-lorraine.fr; 2Department of Molecular Medicine and Personalized Therapeutics, Division of Biochemistry, Molecular Biology, Nutrition, and Metabolism, University Hospital of Nancy, F-54000 Nancy, France; m.julien2@chru-nancy.fr (M.J.); j.levy@chru-nancy.fr (J.L.); o.hajjar@chru-nancy.fr (O.H.); claire.franczak@gmail.com (C.F.); charlotteanne.stephan@gmail.com (C.S.); e.laugel@chru-nancy.fr (E.L.); m.wandzel@chru-nancy.fr (M.W.); pierre.filhine-tresarrieu@hotmail.fr (P.F.-T.); 3Reference Centre for Inborn Errors of Metabolism (ORPHA67872), University Hospital of Nancy, F-54000 Nancy, France; 4Department of Pathology and Laboratory Medicine, University of California, Davis, Sacramento, CA 95817, USA; rgreen@ucdavis.edu

**Keywords:** global health, global burden related to nitrous oxide exposure, medical and recreational settings, individual patient data meta-analysis, nitrous oxide-related toxicity, vitamin B12 deficiency, one-carbon metabolism, homocysteine, methylmalonic acid

## Abstract

The risk of adverse effects of nitrous oxide (N_2_O) exposure is insufficiently recognized despite its widespread use. These effects are mainly reported through case reports. We conducted an individual patient data meta-analysis to assess the prevalence of clinical, laboratory, and magnetic resonance findings in association with N_2_O exposure in medical and recreational settings. We calculated the pooled estimates for the studied outcomes and assessed the potential bias related to population stratification using principal component analysis. Eighty-five publications met the inclusion criteria and reported on 100 patients with a median age of 27 years and 57% of recreational users. The most frequent outcomes were subacute combined degeneration (28%), myelopathy (26%), and generalized demyelinating polyneuropathy (23%). A T2 signal hyperintensity in the spinal cord was reported in 68% (57.2–78.8%) of patients. The most frequent clinical manifestations included paresthesia (80%; 72.0–88.0%), unsteady gait (58%; 48.2–67.8%), and weakness (43%; 33.1–52.9%). At least one hematological abnormality was retrieved in 71.7% (59.9–83.4%) of patients. Most patients had vitamin B12 deficiency: vitamin B12 <150 pmol/L (70.7%; 60.7–80.8%), homocysteine >15 µmol/L (90.3%; 79.3–100%), and methylmalonic acid >0.4 µmol/L (93.8%; 80.4–100%). Consistently, 85% of patients exhibited a possibly or probably deficient vitamin B12 status according to the cB12 scoring system. N_2_O can produce severe outcomes, with neurological or hematological disorders in almost all published cases. More than half of them are reported in the setting of recreational use. The N_2_O-related burden is dominated by vitamin B12 deficiency. This highlights the need to evaluate whether correcting B12 deficiency would prevent N_2_O-related toxicity, particularly in countries with a high prevalence of B12 deficiency.

## 1. Introduction

Nitrous oxide (N_2_O) is a colorless, sweet-smelling gas that has been widely used in dental, emergency, and anesthetic practices. The first use of N_2_O as an anesthetic agent was reported on December 1844 by Dr. Horace Wells, an American dentist who demonstrated insensitivity to pain from a dental extraction after N_2_O inhalation [1]. Before proposing the use of N_2_O as an anesthetic agent, Dr. Wells had attended a demonstration by Gardner Quincy Colton regarding the use of N_2_O as an exhilarating or laughing gas [2]. Joseph Priestley, a British scientist, reported the discovery of N_2_O in 1772, and since that time the inhalation of this gas as part of the public entertainment had become commonplace [3]. Owing to its physical and chemical properties N_2_O is also used in the food industry as a mixing, foaming, and propellant for the preparation of whipping cream.

It is well known that exposure to N_2_O is associated with a risk of developing neurological and hematological complications [4]. In 1956, Lassen et al. reported the first description of severe bone-marrow depression after prolonged N_2_O anesthesia for treating patients with tetanus [4,5]. In one patient from this series, the bone-marrow biopsy on the fifth day of N_2_O anesthesia revealed “strikingly megaloblastic” erythropoiesis and changes in granulocytopoiesis that were typical of “pernicious anemia” [5]. Since this first description of a potential link between N_2_O exposure and myelosuppression, the hypothesis of a disorder related to vitamin B12 metabolism was suggested [6]. The effects of N_2_O exposure are mainly reported through case reports. However, the global burden related to N_2_O exposure, notably regarding its effects on one-carbon metabolism, has never been subject to meta-analysis. Thus, we conducted individual patient data meta-analysis to assess the prevalence of clinical, laboratory, and magnetic resonance findings in association with N_2_O exposure in medical and recreational settings.

## 2. Matherials and Methods

### 2.1. Data Sources and Searches

The literature search was conducted using MEDLINE^®^-indexed literature using the PubMed search engine from the National Centre for Biotechnology Information (www.pubmed.gov) (January 1966 to August 2018) using the following full electronic search strategy: ((protoxide[All Fields] AND (“nitrogen”[MeSH Terms] OR “nitrogen”[All Fields])) OR “Nitrous oxide”[All Fields] OR “nitrogen protoxide”[All Fields] OR “laughing gas”[All Fields] OR whippet[All Fields] OR whippets[All Fields]) AND (b12[All Fields] OR “vitamin b12”[All Fields] OR (“vitamin B12”[MeSH Terms] OR “vitamin B12”[All Fields] OR “cobalamin”[All Fields]) OR (“homocysteine”[MeSH Terms] OR “homocysteine”[All Fields]) OR “methylmalonic acid”[All Fields] OR (“methionine”[MeSH Terms] OR “methionine”[All Fields]) OR “folic Acid”[All Fields]). Additional articles were retrieved from primary search references. EndNote X7.8 was used for reference management [7]. The present systematic review was performed in accordance with the MOOSE (Meta-analysis Of Observational Studies in Epidemiology) Statement [8].

### 2.2. Study Selection

We retained a case report in the systematic review if it reported the description of at least one health outcome in relation to previous N_2_O exposure. The exclusion criteria were as follows: non-English language publication; editorial; narrative review; congress abstract; absence of N_2_O exposure; preventive treatment with vitamin B12 before N_2_O exposure; and no reported data on at least one of the following biological parameters before vitamin B12 therapy: hemoglobin, hematocrit, mean corpuscular volume (MCV), serum folate, homocysteine, or methylmalonic acid.

### 2.3. Data Extraction

Three investigators (A.O.; J.L.; J.-L.G.) reviewed the titles and abstracts of all citations identified by the literature search. Eligible articles were reviewed by eight investigators (A.O.; M.J.; J.L.; O.H.; C.F.; C.S.; E.L.; M.W.; P.F.T.). Disagreement in data extraction was resolved by consensus. The following data were extracted using a predefined extraction form structured in 10 domains: Domain #1: Case report characteristics (Author, Year, Country); Domain #2: Patient’s demographics (Age, Gender); Domain #3: Clinical manifestations expressed as binary outcomes (Group 1: Paresthesia of extremities, numbness, or tingling; Quadriparesis or paralysis; Unsteady gait or walking difficulty; Falling, Equilibrium troubles; Weakness; Paraplegia; Hypotonia; Lhermitte sign; Athetoid movement; Ataxia; Seizures; Spasm; Urinary incontinence; Urine retention; Fecal incontinence; Vertigo; Syncope; Polyneuropathy; Bulbar paralysis; Cognitive decline; Impaired memory; Confusion; Disorientation; Lethargy; Neurological deterioration; Group 2: Behavior alteration; Paranoid behavior; Visual hallucination; Agitation; Depression; Suicidal thought; Group 3: Neck pain; Foot pain; Chest pain; Headache; Painful erection; Abdominal pain; Group 4: Constipation, Anorexia, Vomiting; Group 5: Tachypnea; Apnea; Respiratory paralysis; Group 6: Decreased libido; Hyperpigmentation; Domain #4: Laboratory findings expressed as continuous outcomes [hemoglobin (g/L); hematocrit (%); MCV (fL); vitamin B12 (pmol/L); folate (µg/L); homocysteine (µmol/L); methylmalonic acid (µmol/L)]; We calculated the combined indicator of vitamin B12 status score (cB12) by combining vitamin B12, homocysteine and methylmalonic acid according the Fedosov et al. [9]. The cB12 score defines five vitamin B12 states, as follows: 2 = elevated B12 (cB12 ≥ 1.5), 1 = adequate B12 status (cB12: −0.5 to 1.5), −1 = decreased B12 (cB12: −1.5 to −0.5; start B12 supplements), −2 = possibly B12 deficient (cB12: −2.5 to −1.5; start oral B12), −3 = probably B12 deficient (cB12 < 2.5; start B12 injections); Domain #5: Reported diagnoses expressed as binary outcomes (Subacute combined degeneration; Generalized demyelinating polyneuropathy; Myelopathy; Axonal polyneuropathy; Encephalopathy; Recurrent paraparesis; N_2_O-related toxicity; Vitamin B12 deficiency; MTHFR deficiency; No specific diagnosis applied); Domain #7: Setting of N_2_O exposure expressed as binary outcomes (Recreational use; Surgery; Occupational exposure; Pain management; Manipulation under GA; Sleep disturbance; and Munchausen); Domain #9: N_2_O exposure quantification (frequency: short or regular exposure as defined below; duration of N_2_O exposure in years; N_2_O presentation: cartridge, canister, whippets cream bulbs, anesthesia machine for medical use, or anesthesia machine for personal use); Domain #10: Magnetic resonance findings expressed as binary outcomes (presence or absence of a T2 signal hyperintensity in the spinal cord).

### 2.4. Nitrous Oxide Exposure

Regular N_2_O exposure was defined as repeated exposure to N_2_O, especially in a recreational setting, pain management or occupational exposure, with minimum consumption of one cartridge per month. For each patient, we estimated the average number of cartridges consumed per day and the duration of exposure in years. In the recreational setting, N_2_O was commonly available in the form of small pressurized cartridges which can deliver the equivalent of 8 liters of N_2_O gas at standard temperature and pressure (8 g) [3]. We quantified the exposure to N_2_O using the following formula: Amount of N_2_O exposure = (average number of cartridges consumed per day × duration of exposure expressed in years).

### 2.5. Main Outcomes and Measures

The primary outcome of the systematic review was to report on the clinical, laboratory, and magnetic resonance findings of subjects who were exposed to N_2_O in medical and recreational settings. Main clinical findings: Paresthesia in extremities, numbness, tingling; Unsteady gait, walking difficulty; Weakness; Fallings or equilibrium disorders; Lhermitte’s sign; and Ataxia. Laboratory findings: Hemoglobin; Hematocrit; MCV; Vitamin B12; Folate; Homocysteine; and Methylmalonic acid. Magnetic resonance finding: Presence of T2 signal hyperintensity in the spinal cord. We assessed two secondary outcomes: 1) to look for predictors of regular N_2_O exposure; 2) to assess the potential association between N_2_O exposure and health outcomes.

### 2.6. Data Synthesis and Analysis

Categorical variables were summarized as frequency counts and percentages with the 95% confidence interval (95% CI). Quantitative variables were expressed as medians and interquartile range (IQR, 25^th^ and 75^th^ percentiles). We compared proportions using the chi-square test or Fisher’s exact test as appropriate. Medians were compared using the Mann–Whitney *U* test. We assessed correlations using Spearman’s rank correlation coefficient. To derive predictors of short N_2_O exposure, we performed univariate logistic regression analysis on binary and continuous variables, using the “Short N_2_O exposure” item as a dependent variable. When a continuous variable was significantly associated with a short N_2_O exposure, we carried out receiver operating characteristic (ROC) analysis for defining its optimal cut-off using the “Short exposure to N_2_O” item as a classification variable [10]. The optimal cut-off was defined using the Youden index J [11]. Receiver operating characteristic analysis outputs included the area under the ROC curve (AUROC), the 95% CI, and the associated *p*-value using the exact binomial method. In multivariate analysis, all significant items resulting from the univariate logistic regression were integrated into a multivariate logistic regression model using the stepwise method. All variables with *p* < 0.1 were included in the model and variables with *P* <0.05 were retained in the model. Results were shown as odds ratios (ORs) and 95% CI. We assessed model discrimination using ROC analysis and model calibration using the Hosmer and Lemeshow goodness-of-fit test. All statistical analyses were conducted using the SAS^®^ 9.4 platform (Cary, NC, USA) and MedCalc for Windows v16.8.4 (Ostend, Belgium) based on a two-sided type I error with an alpha level of 0.05.

### 2.7. Assessment of Bias

We assessed the potential bias related to population stratification using principal component analysis. We used all the variables related to clinical findings, laboratory findings, reported diagnoses, and N_2_O exposure settings to calculate the ten top eigenvalues. For each patient, the principal components were calculated according to each eigenvalue. We assessed population stratification by visual inspection using two-dimensional and three-dimensional diagrams. The principal component analysis was conducted using SVS 8.8.1 (Golden Helix, Inc. Bozeman, MT, USA).

## 3. Results

### 3.1. Literature Review

The systematic search generated 513 citations of which 132 appeared to be relevant to the systematic review. Of these 132 studies, 47 were not retained by the selection criteria (Appendix A), leaving 85 eligible publications (Figure 1 and Appendix A). All the case reports included in the systematic review reported individual-level data on 100 patients. Among the 100 reports, most originated from North America (n = 51), Western Europe (n = 22), China/Taiwan (n = 12) and Australia/New Zealand (n = 10).

### 3.2. Prevalence of Clinical, Laboratory and Magnetic Resonance Findings in Association with N_2_O Exposure

Among the 100 patients included in the systematic review, the male:female gender distribution was 60:40 and the median age was 27 years (IQR, 22–36; range, 0.4–76.0). Most patients included in the meta-analysis were exposed to N_2_O in the setting of recreational use (57%) or surgery (25%) (Table 1).

Seventy-six percent (76/100) of patients had regular exposure to N_2_O. Among them, recreational use (73.7%, 56/76), occupational exposure (11.8%, 9/76), and pain management (7.9%, 6/76) represented the three main N_2_O exposure modes. Among regular users, the median N_2_O exposure was 18.5 cartridge-years, corresponding to a total amount of 54,020 g of N_2_O. The three most frequently reported diagnoses were subacute combined degeneration (28%), myelopathy (26%), and generalized demyelinating polyneuropathy (23%). A T2 signal hyperintensity in the spinal cord was reported in 68% of patients who underwent magnetic resonance imaging of the spinal cord (Table 2).

At least one neurological symptom was reported in 96% (92.1–99.9%) of patients and included the following clinical manifestations by decreasing order of frequency: paresthesia in the extremities (80%), unsteady gait or walking difficulties (58%), weakness (43%), fallings or equilibrium disorders (24%), Lhermitte’s sign (15%), and ataxia (12%) (Table 3 and Figure 2).

Patients had a high risk of macrocytic anemia with a median MCV of 100 fL (IQR: 94–103) and median values of hemoglobin of 12.8 g/dL (IQR: 10.8–14.2) and 10.7 g/dL (IQR: 8.3–12.4) in males and females, respectively. At least one hematological abnormality was retrieved in 71.7% of cases (59.9–83.4%). The proportions of patients with low hemoglobin level (<13.0 g/dL in men; <12.0 g/dL in women, based on WHO guidelines [12]), low hematocrit level (<39% in men; <36% in nonpregnant women, based on WHO guidelines [12]), and MCV >100 fL were 55.8%, 52.4%, and 41.8%, respectively (Table 4 and Figure 2).

Regarding one-carbon metabolism markers, the median plasma vitamin B12 concentration was low (101 pmol/L, IQR: 74–161) with 70.7% of patients exhibiting a vitamin B12 level <150 pmol/L, considered as the threshold of vitamin B12 deficiency. The median plasma concentrations of homocysteine and methylmalonic acid were 55 µmol/L (IQR, 29–111) and 5.0 µmol/L (IQR: 1.1–6.6), respectively, with a vast majority of patients exhibiting high homocysteine (>15 µmol/L) and methylmalonic acid (>0.4 µmol/L) levels (90.3% and 93.8%, respectively). According to the cB12 scoring system, 90.9% of patients exhibited at least a decreased vitamin B12 status and 84.8% of patients exhibited a possibly or probably deficient vitamin B12 status (Figure 3). The serum folate concentration was in the normal reference range with a median of 12.8 µg/L (IQR, 7.3–14.6) (Table 4).

### 3.3. Secondary Outcomes

#### 3.3.1. Predictors of Short Nitrous Oxide Exposure

##### Univariate Analysis

In univariate logistic regression analysis, among all the variables screened in the systematic review (63 variables: demographics, n = 2; clinical findings, n = 44; laboratory findings, n = 7; magnetic resonance finding, n = 1; reported diagnoses, n = 9) only three were significantly associated with a short N_2_O exposure, namely: age, vitamin B12 concentration, and MCV (Table 5 and Appendix A). In ROC analysis, age, vitamin B12 concentration, and MCV had a significant optimal cut-off in association with a short N_2_O exposure (Table 5). The dichotomized predictors were significantly associated with a short N_2_O exposure in univariate logistic regression: age ≥40 years (OR = 23.33, 95% CI: 6.84–79.61); vitamin B12 ≤74 pmol/L (OR = 6.06, 95% CI: 2.05–17.90), and MCV >100 fL (OR = 9.75, 95% CI: 1.93–49.15) (Table 5).

##### Multivariate Analysis

Among the variables retained in univariate logistic regression, two remained significant in multivariate logistic regression and were independently associated with a short N_2_O exposure, namely: age ≥40 years (OR = 23.95, 95% CI: 3.62–158.61; *p* = 0.001) and vitamin B12 ≤74 pmol/L (OR = 10.57, 95% CI: 1.70–65.90; *p* = 0.01) (Table 5). The multivariate regression model was well-calibrated and exhibited a good discrimination (AUROC = 0.908; 95% CI: 0.795–0.970) and overall model fit (Cox and Snell R^2^ = 0.42; Nagelkerke R^2^ = 0.60; *p* <0.0001).

#### 3.3.2. Association Between the Amount of Nitrous Oxide Exposure and Outcomes

Data regarding the amount of N_2_O exposure was available in 28 patients. In exploratory analyses, the amount of N_2_O exposure was not significantly correlated with any biological variable. Furthermore, the amount of N_2_O exposure was not significantly associated with the most frequently reported diagnoses (subacute combined degeneration, generalized demyelinating polyneuropathy, and myelopathy), the most commonly reported clinical findings (paresthesia in extremities, numbness, tingling; unsteady gait, walking difficulty; weakness; and fallings or equilibrium disorders), or the presence of T2 signal hyperintensity in the spinal cord (Appendix A).

#### 3.3.3. Assessment of Bias

The visual inspection of 2-D and 3-D plots from the principal component analysis, based on the top five eigenvalues, did not reveal any significant population stratification that could be suggestive of a high risk of bias (Appendix A).

## 4. Discussion

The present meta-analysis highlights the potential side-effects related to N_2_O use with a particularly unfavorable risk-benefit ratio for recreational users who represent more than half of the reported subjects. Among patients with N_2_O-related toxicity, neurological or hematological disorders were observed in almost all patients (96%). The N_2_O-related burden is dominated by vitamin B12 deficiency and the associated alterations in one-carbon metabolism markers.

Three-quarter of patients with N_2_O-related toxicity exhibited a low vitamin B12 status (<150 pmol/L). Consistently, 85% of patients exhibited a possibly or probably deficient vitamin B12 status according to the cB12 scoring system. In the human body, vitamin B12 is physiologically active in two forms: 1) methylcobalamin which represents a cofactor for the methyltransferase enzyme 5-methyltetrahydrofolate-homocysteine methyltransferase, also known as methionine synthase (MTR), and 2) adenosylcobalamin which serves as a cofactor for the enzyme methylmalonyl coenzyme A mutase (MMCoAM) [13,14,15,16,17,18,19,20,21,22,23]. A defect in MMCoAM activity leads to an accumulation of methylmalonic acid while a defect in MTR activity leads to a decrease of methionine and an increase of both homocysteine and S-adenosyl-homocysteine as well as methyltetrahydrofolate concentrations [24]. In the present meta-analysis, vitamin B12 deficiency was evidenced by hyperhomocysteinemia >15 µmol/L and methylmalonic acid >0.4 µmol/L in more than 90% of patients with N_2_O-related toxicity. N_2_O exposure is well described as the most effective way to produce vitamin B12 deficiency in animal models [24,25,26,27]. N_2_O induces oxidation of the active cobalt center contained in the vitamin B12 chemical structure. It also induces a displacement of the vitamin B12 molecule from MTR with a parallel loss of MTR activity [24].

Several mechanistic hypotheses have been advanced for explaining the association between N_2_O exposure and the occurrence of clinical and biochemical abnormalities associated with vitamin B12 deficiency, notably subacute combined degeneration of the spinal cord in association with hyperhomocysteinemia and elevated levels of methylmalonic acid. Hathout & El-Saden nicely described these hypotheses in their review paper regarding N_2_O-induced myelopathy [24]. Three main mechanistic hypotheses have been put forward as the knowledge in the field progressed: the alteration of the MMCoAM pathway [24], the alteration of the methylcobalamin-MTR pathway [24,28,29], and more recently, the imbalance between cytokines and growth factors exhibiting myelinotoxic or myelinotrophic effects [30,31,32,33,34]. Early observations initially suggested that the toxicity of N_2_O occurred primarily through the MMCoAM pathway. However, this was challenged by clinical observations in patients with monogenic inherited disorders associated with methylmalonic acidemia who do not develop subacute combined degeneration [24]. The second hypothesis underlying the relationship between N_2_O exposure and neurological complications is the effect of N_2_O on the methylcobalamin-MTR pathway. Indeed, inherited disorders in MTR activity induce a decrease of methionine and a low methyl-donor status with defective methylation and resulting instability of the myelin sheath. However, this hypothesis was hampered by experiments using the B12-deficient fruit bat animal model. These experiments did not show significant alterations in S-adenosyl-methionine and S-adenosyl-homocysteine in the brain and spinal cord after induction of severe vitamin B12 deficient myelopathy following a combination of dietary deprivation and N_2_O exposure [24,28,29]. During the last two decades, much progress has been made in the field of neuroimmunology and have suggested that the pathogenesis of N_2_O-induced myelopathy could be explained by an imbalance between cytokines and growth factors exhibiting myelinotoxic (tumor necrosis factor alpha, sCD40:sCD40L dyad, nerve growth factor) or myelinotrophic (interleukin-6 and epidermal growth factor) effects [30,31,32,33,34].

The present meta-analysis reports that patients with regular N_2_O exposure exhibited high methylmalonic acid concentrations. In an experimental rat model, the exposure to 50% N_2_O and 50% O_2_ mixture during 48 h was associated with a 70% reduction of MTR activity in the liver, kidney, and brain with no significant changes in the activity of MMCoAM [35]. Prolonged exposure to 50% N_2_O induced a further decrease in both MTR and MMCoAM activities (12% and 32% of control values after 33 days, respectively). This experiment demonstrates the time-dependent effect that N_2_O exerts on the reduction of enzymatic activity of MTR and MMCoAM. The effect of N_2_O on MTR is evident from the first hours of exposure, while the effect on the MMCoAM requires longer exposure times [35]. Prolonged exposure to N_2_O causes a decrease in the activity of both MTR and MMCoAM with harmful effects on one-carbon metabolism markers [35].

In the present meta-analysis, age ≥40 years, a vitamin B12 concentration ≤74 pmol/L, and an MCV >100 fL were associated with a short N_2_O exposure, mostly associated with surgery and a more severe clinical picture. These data are in keeping with those of the ENIGMA trial that assessed the effects of N_2_O on patients’ outcome after major surgery [36]. In this multicenter randomized trial, 215 patients undergoing N_2_O-containing general anesthesia were compared with 179 patients undergoing N_2_O-free general anesthesia [36]. The N_2_O group exhibited a significantly increased risk of postoperative hyperhomocysteinemia defined by rising to the 90^th^ percentile of the preoperative homocysteine concentration (>13.5 µmol/L) (OR = 3.91; 95% CI: 1.82–8.40) [36]. Importantly, the occurrence of postoperative hyperhomocysteinemia was significantly associated with an increased risk of complications (risk ratio (RR) = 2.8; 95% CI: 1.4–5.4) and cardiovascular events (RR = 5.1; 95% CI: 3.1–8.5) [36].

The present meta-analysis reports that almost three-quarters of patients regularly exposed to N_2_O were recreational users. The use of N_2_O as a “laughing gas” became popular in the early 1800s during aristocratic laughing gas parties. Over time, the social image of N_2_O as a recreational agent has shifted from a “high-society drug” to a “psychedelic drug”, ending up during the last decade as a "hippie crack drug.” The 2016/17 Crime Survey for England and Wales reported that 2.6% of adults aged 16–59 (around 840,000 people) had used N_2_O in the preceding year [37]. Among the youngest subjects, the prevalence of N_2_O use was dramatically higher with 9.3% of 16–24 year olds reported using N_2_O (males, 11.1%; females, 7.4%) in the preceding year [37]. The Global Drug Survey 2014 (GDS2014), conducted in 17 countries out of 74,864 participants, confirmed the increasing popularity of N_2_O as a recreational drug with a lifetime prevalence of 38.6% and 29.4% in the UK and US, respectively [38]. Among N_2_O users, the reported rates for persistent numbness and accidental injury were 4.3% and 1.2%, respectively [38]. The updated results from the GDS2016, with over 100,000 respondents from over 50 countries, confirmed that 4% of N_2_O users had symptoms of nerve damage [39]. These data highlight the fact that the recreational use of N_2_O represents a significant public health issue. Awareness campaigns among high-risk populations are eagerly needed, particularly for the 16–24 year olds. Most cases of N_2_O-related toxicity were reported in world areas and countries with a low prevalence of B12 deficiency, including North America and Western Europe. Few reports originated from Turkey and Iran, two Middle Eastern countries with an intermediate-to-high prevalence of vitamin B12 deficiency. This highlights the lack of published data that would allow analyzing the risk of N_2_O-related outcomes in exposed subjects from countries with a high prevalence of vitamin B12 deficiency, including India, sub-Saharan Africa, and Mexico (Appendix A).

The medical community, particularly physicians who are involved in the emergency departments, anesthesiologists, and surgeons, should be aware of the magnitude of recreational N_2_O exposure in the general population [3]. The evaluation of a patient with a suspicion of N_2_O-induced toxicity should include a rigorous clinical evaluation including neurological examination, a laboratory evaluation which consists of a complete blood count, vitamin B12, homocysteine, methylmalonic acid, and folate, and MRI studies where indicated. Patients requiring N_2_O-containing anesthesia, particularly those in the setting of elective surgery, should have the same systematic biochemical assessment. In the ENIGMA trial, the vitamin B supplementation before surgery was associated with a significant reduction of the risk for developing post-operative hyperhomocysteinemia (OR = 0.09; 95% CI: 0.03–0.28) [36]. Thus, a strategy based on preoperative screening and the correction of a possible vitamin B12 deficiency, notably in high-risk populations (elderly, vegan, and chronically ill patients) could reduce the risk of N_2_O-related disorders.

Two systematic reviews assessed N_2_O-related toxicity in recreational use and general anesthesia settings. Garakani et al. reported the main findings related to chronic N_2_O abuse with a focus on neurological sequelae and psychiatric disorders [4]. In this systematic review, no summary effect was calculated regarding medical, laboratory, and radiological findings. Furthermore, biological findings were assessed as categorical variables, and no univariate or multivariate analyses were performed. The authors concluded that chronic N_2_O abuse represents a potentially difficult-to-diagnose condition which could lead to death and that physicians should be aware of N_2_O-related toxicity. In the setting of anesthesia, a meta-analysis summarized the evidence from randomized clinical trials associating N_2_O with serious cardiovascular complications [40]. This meta-analysis concluded on the lack of robust evidence for how N_2_O used as part of general anesthesia affects mortality and cardiovascular complications [40].

The present meta-analysis has several strengths. First, we report an individual patient data meta-analysis that collected original data from 100 patients to describe the global burden related to N_2_O toxicity. This approach has also allowed performing univariate and multivariate analyses. Second, the present meta-analysis reported quantitative evidence about the status of one-carbon metabolism markers in the setting of N_2_O-related toxicity and highlighted the risk of hyperhomocysteinemia and high methylmalonic acidemia. Third, the meta-analysis of case reports allowed the compilation of unselected patients, thereby reducing the risk of population heterogeneity. The analysis of bias did not reveal a significant stratification of the analyzed population. We acknowledge several limitations. First, we used data extracted from available case reports through a systematic retrospective search. This strategy carries the risk of missing data. Second, laboratory findings were not available for all patients, which resulted in a decrease in the statistical power for univariate and multivariate analyses. Third, in the surgical setting, thromboembolic events potentially associated with N_2_O-induced hyperhomocysteinemia could have been mistakenly attributed to the postoperative context, leading to a potential underestimation of the thromboembolic risk associated with N_2_O exposure [36,41,42,43,44,45,46,47]. Fourth, in the present systematic review, the exclusion of non-English case reports could have potentially led to selection bias. However, a study that examined the influence of non-English publications on combined estimates of published meta-analyses did not reveal a significant effect after the exclusion of non-English publications [48]. Fifth, because our meta-analysis focuses on single patient reports or small case series, we cannot evaluate the frequency of events related to N_2_O-induced toxicity. These events may be relatively rare as N_2_O is widely used. Our data point out the need to perform large population studies to better report on N_2_O-induced toxicity.

## 5. Conclusions

This meta-analysis points out the association of N_2_O exposure with severe neurological and hematological manifestations and altered one-carbon metabolism markers. These findings lead to raising awareness among the medical community urgently about the risk of N_2_O exposure and particularly in the recreational setting among young people. Population studies are warranted to evaluate whether the correction of vitamin B12 deficiency prevents N_2_O-related toxicity in the context of anesthesia and recreational use, particularly in countries with a high prevalence of vitamin B12 deficiency.

## Figures and Tables

**Figure 1 jcm-08-00551-f001:**
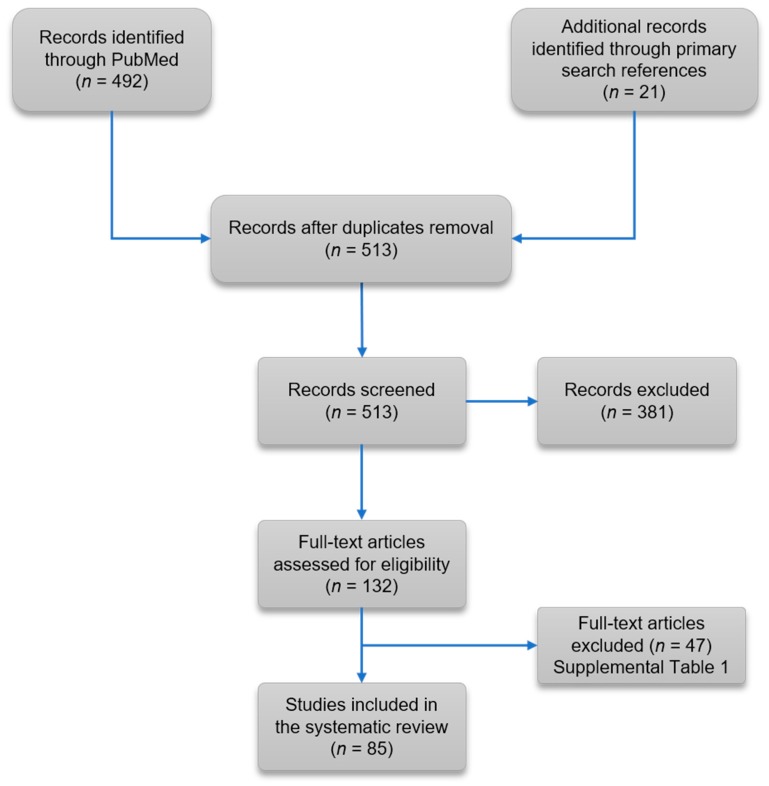
Flow diagram of the systematic review.

**Figure 2 jcm-08-00551-f002:**
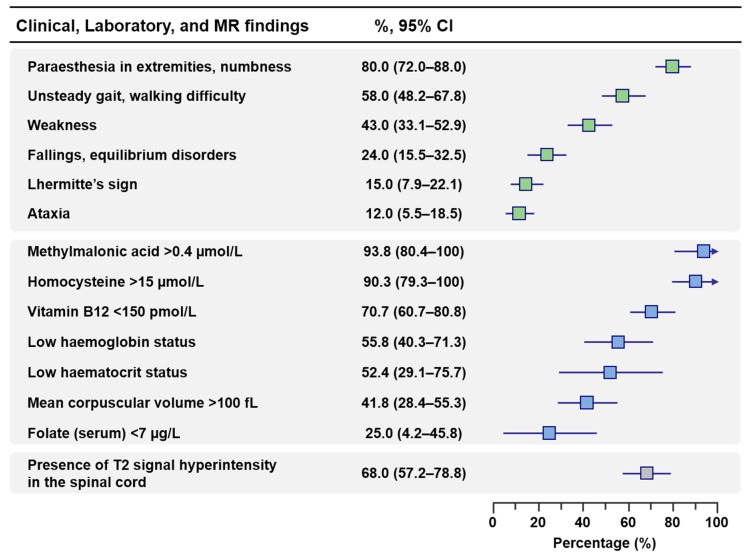
Proportion (95% confidence interval) of main clinical, laboratory and radiological findings in patients exposed to nitrous oxide (N_2_O). Low hemoglobin status was defined according to WHO guidelines (<13.0 g/dL in men; <12.0 g/dL in nonpregnant women) [12]. Low hematocrit status was defined according to WHO guidelines (<39% in men; <36% in nonpregnant women) [12].

**Figure 3 jcm-08-00551-f003:**
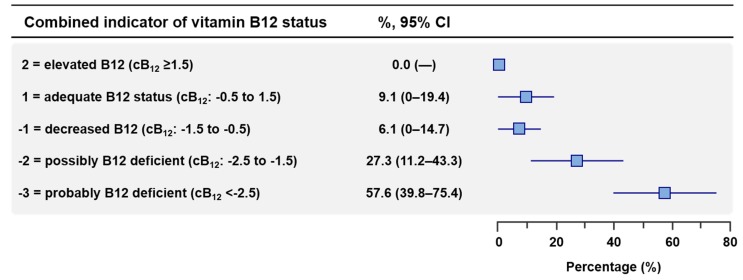
Vitamin B12 status according to the combined indicator of vitamin B12 status (cB12) scoring system. The cB12 score was calculated according to Fedosov et al. [9]. Proportion (95% confidence interval) of main clinical, laboratory and radiological findings in patients exposed to nitrous oxide (N_2_O). Low hemoglobin status was defined according to WHO guidelines (<13.0 g/dL in men; <12.0 g/dL in nonpregnant women) [12]. Low hematocrit status was defined according to WHO guidelines (<39% in men; <36% in nonpregnant women) [12].

**Table 1 jcm-08-00551-t001:** Settings and Quantification of Nitrous Oxide Exposure Among Patients Included in the Individual Patient Data Meta-Analysis.

**Setting of nitrous oxide exposure**	**N**	**n**	**Percentage (95% CI)**
Recreational	100	57	57.0 (47.1–66.9)
Surgery	100	25	25.0 (16.4–33.6)
Occupational exposure	100	9	9.0 (3.3–14.7)
Pain management	100	6	6.0 (1.3–10.7)
Manipulation under general anesthesia	100	1	1.0 (0–3.0)
Munchausen syndrome	100	1	1.0 (0–3.0)
Management of sleep disturbance	100	1	1.0 (0–3.0)
**Frequency of nitrous oxide exposure**	**N**	**n**	**Percentage (95% CI)**
Regular	100	76	76.0 (67.5–84.5)
Once	100	24	24.0 (15.5–32.5)
**Quantitative estimates of nitrous oxide exposure***	**n**	**Median**	**IQR (25^th^–75^th^)**
Number of nitrous oxide cartridge per day	30	25	8–85
Duration of nitrous oxide duration (year)	52	0.7	0.3–1.5
Quantification of nitrous oxide exposure (cartridge-years)	28	18.5	1.4–99.9

IQR: interquartile range; N: number of studied patients; n: number of observations. * Among patients with regular nitrous oxide exposure.

**Table 2 jcm-08-00551-t002:** Magnetic Resonance Findings and Diagnoses Among the Patients Included in the Individual Patient Data Meta-Analysis.

**Magnetic resonance imaging findings**	**N**	**n**	**Percentage (95% CI)**
Presence of T2 signal hyperintensity in the spinal cord	75	51	68.0 (57.2–78.8)
**Reported diagnoses***	**N**	**n**	**Percentage (95% CI)**
Subacute combined degeneration	100	28	28.0 (19.0–37.0)
Myelopathy	100	26	26.0 (17.3–34.7)
Generalized demyelinating polyneuropathy	100	23	23.0 (14.6–31.4)
Vitamin B12 deficiency	100	14	14.0 (7.1–20.9)
Axonal polyneuropathy	100	11	11.0 (4.8–17.2)
Encephalopathy	100	2	2.0 (0–4.8)
Recurrent paraparesis	100	1	1.0 (0–3)
MTHFR deficiency	100	1	1.0 (0–3)
Toxicity due to N_2_O with no specific diagnosis applied	100	19	19.0 (11.2–26.8)

N: total number of studied patients; n: number of observations; N_2_O: nitrous oxide. * Any patient could have more than one diagnosis applied.

**Table 3 jcm-08-00551-t003:** Clinical Findings Among the Patients Included in the Individual Patient Data Meta-Analysis.

Clinical Findings	N	n	Percentage (95% CI)
Paresthesia in extremities, numbness, tingling	100	80	80.0 (72.0–88.0)
Unsteady gait, walking difficulty	100	58	58.0 (48.2–67.8)
Weakness	100	43	43.0 (33.1–52.9)
Fallings or equilibrium disorders	100	24	24.0 (15.5–32.5)
Lhermitte’s sign	100	15	15.0 (7.9–22.1)
Ataxia	100	12	12.0 (5.5–18.5)
Cognitive decline	100	9	9.0 (3.3–14.7)
Urinary incontinence	100	8	8.0 (2.6–13.4)
Quadriparesis or paralysis	100	7	7.0 (1.9–12.1)
Behavior alteration	100	6	6.0 (1.3–10.7)
Urine retention	100	5	5.0 (0.7–9.4)
Impaired memory	100	5	5.0 (0.7–9.4)
Headache	100	4	4.0 (0.1–7.9)
Depression	100	4	4.0 (0.1–7.9)
Thrombo-occlusive event	100	3	3.0 (0–6.4)
Mental confusion	100	3	3.0 (0–6.4)
Constipation	100	3	3.0 (0–6.4)
Paranoid behavior	100	3	3.0 (0–6.4)
Foot pain	100	3	3.0 (0–6.4)
Hyperpigmentation	100	3	3.0 (0–6.4)
Abdominal pain	100	3	3.0 (0–6.4)
Agitation	100	2	2.0 (0–4.8)
Fecal incontinence	100	2	2.0 (0–4.8)
Lethargy	100	2	2.0 (0–4.8)
Seizures	100	2	2.0 (0–4.8)
Decreased libido	100	2	2.0 (0–4.8)
Visual hallucination	100	2	2.0 (0–4.8)
Anorexia	100	1	1.0 (0–3)
Apnea	100	1	1.0 (0–3)
Athetoid movement	100	1	1.0 (0–3)
Bulbar paralysis	100	1	1.0 (0–3)
Chest pain	100	1	1.0 (0–3)
Disorientation	100	1	1.0 (0–3)
Hypotonia	100	1	1.0 (0–3)
Neck pain	100	1	1.0 (0–3)
Painful erection	100	1	1.0 (0–3)
Paraplegia	100	1	1.0 (0–3)
Polyneuropathy	100	1	1.0 (0–3)
Respiratory paralysis	100	1	1.0 (0–3)
Spasm	100	1	1.0 (0–3)
Suicidal thought	100	1	1.0 (0–3)
Syncope	100	1	1.0 (0–3)
Tachypnea	100	1	1.0 (0–3)
Vertigo	100	1	1.0 (0–3)
Vomiting	100	1	1.0 (0–3)

N: total number of studied patients; n: number of observations.

**Table 4 jcm-08-00551-t004:** Laboratory Findings Among the Patients Included in the Individual Patient Data Meta-Analysis.

**Laboratory findings (continuous)**	**n**	**Median**	**IQR (25^th^–75^th^)**
Hemoglobin (g/dL)*	43	12.0	9.1–13.3
*Males*	23	12.8	10.8–14.2
*Females*	20	10.7	8.3–12.4
Hematocrit (%)†	21	38	33–42
*Males*	13	40	33–44
*Females*	8	35	32–39
Mean corpuscular volume (fL)	55	100	94–103
Vitamin B12 (pmol/L)	82	101	74–161
Folate (serum) (µg/L)	20	12.8	7.3–14.6
Homocysteine (µmol/L)	31	55	29–111
Methylmalonic acid (µmol/L)	16	5.0	1.1–6.6
Combined indicator of vitamin B12 status	33	−2.802	−3.368–−1.924
**Laboratory findings (dichotomized)**	**N**	**n**	**Percentage (95%, CI)**
Low hemoglobin status*	43	24	55.8 (40.3–71.3)
Low hematocrit status†	21	21	52.4 (29.1–75.7)
Mean corpuscular volume >100 fL	55	23	41.8 (28.4–55.3)
Vitamin B12 < 150 pmol/L	82	58	70.7 (60.7–80.8)
Folate (serum) <7 µg/L	20	5	25.0 (4.2–45.8)
Homocysteine >15 µmol/L	31	28	90.3 (79.3–100)
Methylmalonic acid >0.4 µmol/L	16	15	93.8 (80.4–100)
**Combined indicator of vitamin B12 status‡**			
2 = elevated B12 (cB12 ≥ 1.5)	33	0	0.0 (—)
1 = adequate B12 status (cB12: −0.5 to 1.5)	33	3	9.1 (0–19.4)
−1 = decreased B12 (cB12: −1.5 to -0.5)	33	2	6.1 (0–14.7)
−2 = possibly B12 deficient (cB12: −2.5 to −1.5)	33	9	27.3 (11.2–43.3)
−3 = probably B12 deficient (cB12 < −2.5)	33	19	57.6 (39.8–75.4)

IQR: interquartile range; N: total number of studied patients; n: number of observations. *Low hemoglobin status was defined according to WHO guidelines (<13.0 g/dL in men; <12.0 g/dL in nonpregnant women) [12]. † Low hematocrit status was defined according to WHO guidelines (<39% in men; <36% in nonpregnant women) [12]. ‡ The combined indicator of vitamin B12 status (cB12) score was calculated according to Fedosov et al. [9].

**Table 5 jcm-08-00551-t005:** Factors Associated with a Short Exposure to Nitrous Oxide in Univariate and Multivariate Analyses.

Predictor	Short Exposure to N_2_O, Median (IQR)	Regular Exposure to N_2_O, Median (IQR)	AUROC† Defined Cut-Off	AUROC, *p*-Value	Univariate LR*, OR (95% CI)	Univariate LR*, *p*-Value	Multivariate LR‡, OR (95% CI)	Multivariate LR‡, *p*-Value
Age (years)	47 (25–58)	26 (22–33)	≥40	0.0076	23.33 (6.84–79.61)	< 0.0001	23.95 (3.62–158.61)	0.001
Vitamin B12 (pmol/L)	74 (33–104)	110 (81–194)	≤74	0.0002	6.06 (2.05–17.90)	0.001	10.57 (1.70–65.90)	0.01
MCV (fL)	104 (101–110)	97 (92–101)	>100	<0.0001	9.75 (1.93–49.15)	0.006	Not retained§	Not retained§

95% CI: 95% confidence interval; AUROC: area under the receiver operating characteristic curve; IQR: interquartile range; LR: logistic regression; MCV: mean corpuscular volume; OR: odds ratio. * Univariate logistic regression analysis on dichotomized variables; † Classification variable: Short nitrous oxide exposure; ‡ Multivariate logistic regression analysis on dichotomized variables using the stepwise method; § Not retained in the multivariate logistic regression model.

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
