# Peer review of "Global Burden Related to Nitrous Oxide Exposure in Medical and Recreational Settings: A Systematic Review and Individual Patient Data Meta-Analysis"

_jcm, 2019, doi:10.3390/jcm8040551_

Reviewer 1 Report

As the title describes, the paper discusses the impact of N2O use in the medical and recreational use setting by forming a systematic review of cases. Overall the paper is very well written and elaborate. I have some comments:

Generally

- While I agree with the authors that this association is little known and requires more recognition, the statement that N2O is more dangerous than previously thought needs to be attenuated a little bit. This is because the paper focuses on small or single patient reports and therefore, these events are rare, as N2O is widely used. If there was indeed great danger there would presumably be studies larger than case reports reporting these events. 

- There are 2 previous systematic reviews that have been done on the subject: one on patients with recreational use, and another on patients in the setting of anesthesia. The authors should therefore discuss this and state the purpose and objective of combining these combinations. To me these are slightly different patient groups, and this was validated by your work. 

Methods

- I may have missed it but the authors should indicate when their search was done. 

Limitations

- Given the rarity of the cases it would have been useful to include foreign language articles as this could have potentially increased your overall n. This should be acknowledged as a limitation.

Author Response

Manuscript ID: jcm-480322

Point-by-point response

REVIEWER #1

Comment #1. English language and style: English language and style are fine/minor spell check required 

Response #1. We thank Reviewer #1 for her/his comment.

***

Comment #2. As the title describes, the paper discusses the impact of N2O use in the medical and recreational use setting by forming a systematic review of cases. Overall the paper is very well written and elaborate

Response #1. We thank Reviewer #1 for her/his insightful comments that have improved the manuscript.

***

Comment #3. While I agree with the authors that this association is little known and requires more recognition, the statement that N2O is more dangerous than previously thought needs to be attenuated a little bit. This is because the paper focuses on small or single patient reports and therefore, these events are rare, as N2O is widely used. If there was indeed great danger, there would presumably be studies larger than case reports reporting these events.

Answer #3. We agree with Reviewer #1. We have now toned down the conclusion of the systematic review in the paragraph on strengths and limitations as follows: “Fifth, because our meta-analysis focuses on single patient reports or small case series, we cannot evaluate the frequency of events related to N2O-induced toxicity. These events may be relatively rare as N2O is widely used. Our data point out the need to perform large population studies to better report on N2O-induced toxicity events.”

***

Comment #4. There are 2 previous systematic reviews that have been done on the subject: one on patients with recreational use, and another on patients in the setting of anesthesia. The authors should therefore discuss this and state the purpose and objective of combining these combinations. To me these are slightly different patient groups, and this was validated by your work. 

Answer #4. We agree with Reviewer #1. We have now discussed the results from two previously published systematic reviews concerning N2O-induced toxicity, as follows: “Two systematic reviews assessed N2O-related toxicity in recreational use and general anesthesia settings. Garakani et al. reported the main findings related to chronic N2O abuse with a focus on neurological sequelae and psychiatric disorders [4]. In this systematic review, no summary effect was calculated regarding medical, laboratory, and radiological findings. Furthermore, biological findings were assessed as categorical variables, and no univariate or multivariate analyses were performed. The authors concluded that chronic N2O abuse represents a potentially difficult-to-diagnose condition which could lead to death and that physicians should be aware of N2O-related toxicity. In the setting of anesthesia, a meta-analysis summarized the evidence from randomized clinical trials associating N2O with serious cardiovascular complications [40]. This meta-analysis concluded on the lack of robust evidence for how N2O used as part of general anesthesia affects mortality and cardiovascular complications [40].”.

***

Comment #5. I may have missed it, but the authors should indicate when their search was done. 

Answer #5. As reported in the Methods section, the last update of the systematic review was performed on August 2018: “The literature search was conducted using MEDLINE®-indexed literature using the PubMed search engine from the National Centre for Biotechnology Information (www.pubmed.gov) (January 1966 to August 2018) using the following full electronic search strategy”.

***

Comment #6. Given the rarity of the cases it would have been useful to include foreign language articles as this could have potentially increased your overall. This should be acknowledged as a limitation.

Answer #6. We agree with Reviewer #1. We have highlighted this limitation in the Discussion section of the revised manuscript, as follows: “Fourth, in the present systematic review, the exclusion of non-English case reports could have potentially led to selection bias. However, a study that examined the influence of non-English publications on combined estimates of published meta-analyses did not reveal a significant effect after the exclusion of non-English publications [46].”

***

Reviewer 2 Report

General comments

The manuscript is a very well written work, which summarizes and analyzes the data from literature about N2O effect on B12-dependent metabolism and physiological manifestations of B12 deficiency. The paper can be accepted after a minor revision. I suggest a few optional modifications listed below, because I think that their incorporation would benefit the manuscript, see Specific comments.

Specific comments

(1) Figure 2, Table 4 and the accompanying text. I miss presentation of the data in terms of the combined diagnostics of B12 status, which is steadily growing in its popularity after public release of the calculation spreadsheets for 2, 3 and 4 markers, see Fedosov et al. (2015) Combined indicator of vitamin B12 status: modification for missing biomarkers and folate status and recommendations for revised cut-points. Clin. Chem. Lab. Med. 53, 1215–1225; doi: 10.1515/cclm-2014-0818. This is interesting, how the combined index behaves during exposure of patients and volunteers to N2O.

(2) Table 5. I think that presentation and discussion of Table 5 (Factors Associated with Regular Exposure to Nitrous Oxide in Univariate and Multivariate Analyses) needs some adjustment. The same is true for the related Supplementary Table 3. The current layout of both tables creates impression that a regular inhalation of N2O is beneficial for B12 status (in comparison to a single exposure). The authors should stress that a single exposure (apparently in course of a surgery) involves a much larger dose of N2O than a continuous recreational use. In such way, the impact on B12 status would expectedly be more pronounced after surgery. If my inference is not true, the material also needs further clarification, as typical readers (like myself) might be confused.

(3) Table 5 and Supplementary Table 3 with the related main text. The authors do not include homocysteine in the list of significant factors during comparison of Single vs Regular exposure to N2O. This decision is formally accurate, as p = 0.06 is higher than the critical value of p = 0.05. Yet, such formal interpretation leads to a rather odd situation, e.g. p = 0.0501 is insignificant while p = 0.0499 is significant. I suggest to add homocysteine values to Table 3 with mentioning that this parameter does not pass the criterion of p = 0.05 but is very much close to it and a larger cohort might have caused incorporation of homocysteine into this list. Application of another nonparametric test (e.g. Kolmogorov–Smirnov instead of Mann‐Whitney test) might also give a subject for speculations. For example Kolmogorov–Smirnov test is more sensitive, when a part of distribution “A” considerably deviates from distribution “B”, while medians are sufficiently close.

(4) I miss discussion about the mechanistic effect of N2O on B12 and its metabolism, because this is not a trivial matter. Some information is provided in Chanarin (1980) Cobalamins and nitrous oxide: a review. J Clin Pathol 33:909-916, where the following reactions are suggested:

Cbl(I) + N20 → Cbl(III) + H20 + N2

Cbl(III) + Cbl(I) → 2·Cbl(II)

Yet, Cbl(II) is a “normal” metabolic product, generated under processing of all freshly incoming Cbl forms. Based on Pratt (1972) Inorganic chemistry of vitamin B12, Academic press, chapter 11.V (p.204) and chapter 15.II, one can deduce that Cbl(II) binds O2 with the following generation of reactive oxygen species (HO2- and H2O2), which in turn oxidize Cbl(II), giving a considerable fraction of inactive yellow corrinoids. If the authors found any useful information under their investigation of the literature, they are very welcome to share it with the readers.

Minor comments

Line 38. Change the text to “… and propellant for the preparation of whipping cream.”

Line 386. Change the text to “… were not available for all patients …”

Author Response

Manuscript ID: jcm-480322

 Point-by-point response

 REVIEWER #2

Comment #1. English language and style: English language and style are fine/minor spell check required.

Response #1. We thank Reviewer #2 for her/his comment.

***

 Comment #2. The manuscript is a very well written work, which summarizes and analyzes the data from literature about N2O effect on B12-dependent metabolism and physiological manifestations of B12 deficiency. The paper can be accepted after a minor revision. I suggest a few optional modifications listed below, because I think that their incorporation would benefit the manuscript, see Specific comments.

Answer #2. We thank Reviewer #2 for her/his insightful comments that have further improved the manuscript.

***

 Comment #3. Figure 2, Table 4 and the accompanying text. I miss presentation of the data in terms of the combined diagnostics of B12 status, which is steadily growing in its popularity after public release of the calculation spreadsheets for 2, 3 and 4 markers, see Fedosov et al. (2015) Combined indicator of vitamin B12 status: modification for missing biomarkers and folate status and recommendations for revised cut-points. Clin. Chem. Lab. Med. 53, 1215–1225; doi: 10.1515/cclm-2014-0818. This is interesting, how the combined index behaves during exposure of patients and volunteers to N2O.

Answer #3. We thank Reviewer #1 for her/his comment. We have now calculated the combined indicator of vitamin B12 status cB12 using the publicly available calculation spreadsheets. We calculated the 3cB12 for 13 patients using vitamin B12, homocysteine, and methylmalonic acid; and the 2cB12 score for 20 patients using the vitamin B12 with either homocysteine (n=17) or methylmalonic acid (n=3). We have updated the manuscript using the cB12 scoring system as follows:

Abstract

“Most patients had vitamin B12 deficiency: vitamin B12<150 pmol/L (70.7%; 60.7%–80.8%), homocysteine >15 µmol/L (90.3%; 79.3%–100%), and methylmalonic acid >0.4 µmol/L (93.8%; 80.4%–100%). Consistently, 85% of patients exhibited a possibly or probably deficient vitamin B12 status according to the cB12 scoring system.”

Methods:

“Domain #4: Laboratory findings expressed as continuous outcomes [hemoglobin (g/L); hematocrit (%); MCV (fL); vitamin B12 (pmol/L); folate (µg/L); homocysteine (µmol/L); methylmalonic acid (µmol/L)]; We calculated the combined indicator of vitamin B12 status score (cB12) by combining vitamin B12, homocysteine and methylmalonic acid according the Fedosov et al. [9]. The cB12 score defines five vitamin B12 states, as follows: 2 = elevated B12 (cB12 ≥1.5), 1 = adequate B12 status (cB12: -0.5 to 1.5), -1 = decreased B12 (cB12: -1.5 to -0.5; start B12 supplements), -2 = possibly B12 deficient (cB12: -2.5 to -1.5; start oral B12), -3 = probably B12 deficient (cB12 <2.5; start B12 injections)”

Results

According to the cB12 scoring system, 90.9% of patients exhibited at least a decreased vitamin B12 status and 84.8% of patients exhibited a possibly or probably deficient vitamin B12 status (Figure 3).”

Tables

Table 4. Laboratory Findings Among the Patients Included in the Individual Patient Data Meta-Analysis

Laboratory   findings (continuous)

n

Median

IQR (25th – 75th)

Hemoglobin   (g/dL)*

43

12.0

9.1–13.3

Males

23

12.8

10.8–14.2

Females

20

10.7

8.3–12.4

Hematocrit   (%)†

21

38

33–42

Males

13

40

33–44

Females

8

35

32–39

Mean   corpuscular volume (fL)

55

100

94–103

Vitamin   B12 (pmol/L)

82

101

74–161

Folate   (serum) (µg/L)

20

12.8

7.3–14.6

Homocysteine   (µmol/L)

31

55

29–111

Methylmalonic   acid (µmol/L)

16

5.0

1.1–6.6

Combined indicator of   vitamin B12 status

33

-2.802

-4.183–0.510

Laboratory findings (dichotomized)

N

n

Percentage (95%, CI)

Low   hemoglobin status*

43

24

55.8 (40.3–71.3)

Low   hematocrit status†

21

21

52.4 (29.1–75.7)

Mean   corpuscular volume >100 fL

55

23

41.8 (28.4–55.3)

Vitamin   B12<150 pmol/L

82

58

70.7 (60.7–80.8)

Folate   (serum)<7 µg/L

20

5

25.0 (4.2–45.8)

Homocysteine   >15 µmol/L

31

28

90.3 (79.3–100)

Methylmalonic   acid >0.4 µmol/L

16

15

93.8 (80.4–100)

Combined indicator of vitamin B12 status‡

 2 = elevated B12 (cB12 ≥ 1.5)

33

0

0.0 (—)

 1 = adequate B12 status (cB12:   -0.5 to 1.5)

33

3

9.1 (0–19.4)

-1 = decreased B12 (cB12:   -1.5 to -0.5)

33

2

6.1 (0–14.7)

-2 = possibly B12 deficient   (cB12: -2.5 to -1.5)

33

9

27.3 (11.2–43.3)

-3 = probably B12 deficient   (cB12 < -2.5)

33

19

57.6 (39.8–75.4)

IQR: interquartile range; N: total number of studied patients; n: number of observations.

* Low hemoglobin status was defined according to WHO guidelines (<13.0 g/dL in men; <12.0 g/dL in nonpregnant women) [12].

† Low hematocrit status was defined according to WHO guidelines (<39% in men; <36% in nonpregnant women) [12].

‡ The combined indicator of vitamin B12 status (cB12) score was calculated according to Fedosov et al. [9]

Figures

Figure 3. Vitamin B12 status according to the combined indicator of vitamin B12 status (cB12) scoring system. The cB12 score was calculated according to Fedosov et al. [9]

Discussion

Three-quarter of patients with N2O-related toxicity exhibited a low vitamin B12 status (<150 pmol/L). Consistently, 85% of patients exhibited a possibly or probably deficient vitamin B12 status according to the cB12 scoring system.

***

Comment #4. Table 5. I think that presentation and discussion of Table 5 (Factors Associated with Regular Exposure to Nitrous Oxide in Univariate and Multivariate Analyses) needs some adjustment. The same is true for the related Supplementary Table 3. The current layout of both tables creates impression that a regular inhalation of N2O is beneficial for B12 status (in comparison to a single exposure). The authors should stress that a single exposure (apparently in course of a surgery) involves a much larger dose of N2O than a continuous recreational use. In such way, the impact on B12 status would expectedly be more pronounced after surgery. If my inference is not true, the material also needs further clarification, as typical readers (like myself) might be confused.

Answer #4. We agree with Reviewer #1. We have now updated the manuscript according to the Reviewer’s comment. All univariate and multivariate analyses are reported for the association with a short exposure to nitrous oxide. By this way, the presentation will be more intuitive and will help the reader understanding that a single exposure, which is notably reported in the setting of surgery, is associated with a worse outcome regarding vitamin B12, MCV. We have also updated Table 5 accordingly. In the Discussion section we have stressed that patients with short exposition have a more severe clinical picture, as follows: “In the present meta-analysis, age ≥ 40 years, a vitamin B12 concentration ≤ 74 pmol/L, and an MCV >100 fL were associated with a short N2O exposure, mostly associated with surgery and a more severe clinical picture. These data are in keeping with those of the ENIGMA trial that assessed the effects of N2O on patients’ outcome after major surgery [35]. In this multicenter randomized trial, 215 patients undergoing N2O-containing general anesthesia were compared with 179 patients undergoing N2O-free general anesthesia [35]. The N2O group exhibited a significantly increased risk of postoperative hyperhomocysteinemia defined by rising to the 90th percentile of the preoperative homocysteine concentration (>13.5 µmol/L) (OR = 3.91; 95% CI: 1.82–8.40) [35]. Importantly, the occurrence of postoperative hyperhomocysteinemia was significantly associated with an increased risk of complications (risk ratio [RR] = 2.8; 95% CI: 1.4–5.4) and cardiovascular events (RR = 5.1; 95% CI: 3.1–8.5) [35].”

***

Comment #5. Table 5 and Supplementary Table 3 with the related main text. The authors do not include homocysteine in the list of significant factors during comparison of Single vs Regular exposure to N2O. This decision is formally accurate, as p = 0.06 is higher than the critical value of p = 0.05. Yet, such formal interpretation leads to a rather odd situation, e.g. p = 0.0501 is insignificant while p = 0.0499 is significant. I suggest to add homocysteine values to Table 3 with mentioning that this parameter does not pass the criterion of p = 0.05 but is very much close to it and a larger cohort might have caused incorporation of homocysteine into this list. Application of another nonparametric test (e.g. Kolmogorov–Smirnov instead of Mann‐Whitney test) might also give a subject for speculations. For example Kolmogorov–Smirnov test is more sensitive, when a part of distribution “A” considerably deviates from distribution “B”, while medians are sufficiently close.

Answer #5. We agree with Reviewer #1 that the P-value associated with the ‘homocysteine’ variable in the univariate analysis is borderline (P=0.06), suggesting its potential use in multivariate analysis. The multivariate model that includes ‘age’, ‘vitamin B12’, and ‘MCV’ as predictors reports on a sample size of 52 patients (the log file of the multivariate regression model is available below). When the ‘homocysteine’ variable is added to the model, the number of subjects drops to 21, because of missing data, which precludes a robust multivariate logistic regression analysis. Furthermore, the ‘vitamin B12’ and ‘homocysteine’ variables are highly and significantly correlated with each other (Spearman's coefficient of rank correlation = -0.525; 95% CI: -0.744 to -0.203; P=0.003) (See Figure 1 below). Thus, the combination of ‘vitamin B12’ and ‘homocysteine’ in the same multivariate model as predictors will introduce collinearity which can cause unstable estimates and inaccurate variances which affects confidence intervals and hypothesis testing. For these two main reasons (sample size shrinking after adding ‘homocysteine’ and collinearity between B12 and homocysteine), we did not use the ‘homocysteine’ variable as a potential predictor in the multivariate logistic regression model.

Log file of the multivariate logistic regression model

Dependent Y

Short N2O exposure

Method

Stepwise

Enter variable if P<< p="">

0.05

Remove variable if P>

0.1

Sample size

52

Positive cases a

14 (26.92%)

Negative cases b

38 (73.08%)

a Short N2O exposure
b Regular N2O exposure

Overall Model Fit

Null model -2 Log Likelihood

60.579

Full model -2 Log Likelihood

32.687

Chi-squared

27.892

DF

2

Significance level

P < 0.0001

Cox & Snell R2

0.4151

Nagelkerke R2

0.6033

Coefficients and Standard Errors

Variable

Coefficient

Std. Error

Wald

P

Age>=40

3.17613

0.96445

10.8453

0.0010

Bio_B12__ng_L_<=100< p="">

2.35849

0.93346

6.3837

0.0115

Constant

-3.07303

0.78237

15.4279

0.0001

Variables not retained in the model

Bio_MCV>100

Odds Ratios and 95% Confidence Intervals

Variable

Odds ratio

95% CI

Age>=40

23.9539

3.6176 to 158.6096

Bio_B12__ng_L_<=100< p="">

10.5749

1.6971 to 65.8958

Hosmer & Lemeshow test

Chi-squared

?

DF

3

Significance level

P = 1.0000

Contingency table for Hosmer & Lemeshow test [Show]

Contingency table for Hosmer & Lemeshow test

Group

Y=0

Y=1

Total

Observed

Expected

Observed

Expected

1

0

0.000

0

0.000

0

2

30

29.629

1

1.371

31

3

5

5.371

3

2.629

8

4

2

2.371

3

2.629

5

5

1

0.629

7

7.371

8

Classification table (cut-off value p=0.5)

Actual group

Predicted group

Percent correct

0

1

Y = 0        

35

3

92.11%

Y = 1        

4

10

71.43%

Percent of cases correctly classified

86.54%

ROC curve analysis

Area under the ROC curve (AUC) 

0.908

Standard Error

0.0505

95% Confidence interval

0.795 to 0.970

Figure 1. Correlation between vitamin B12 and Homocysteine. The statistical significance was assessed using the Spearman’s rank correlation coefficient.

***

Comment #6. I miss discussion about the mechanistic effect of N2O on B12 and its metabolism, because this is not a trivial matter. Some information is provided in Chanarin (1980) Cobalamins and nitrous oxide: a review. J Clin Pathol 33:909-916, where the following reactions are suggested:

Cbl(I) + N20 → Cbl(III) + H20 + N2

Cbl(III) + Cbl(I) → 2·Cbl(II)

Yet, Cbl(II) is a “normal” metabolic product, generated under processing of all freshly incoming Cbl forms. Based on Pratt (1972) Inorganic chemistry of vitamin B12, Academic press, chapter 11.V (p.204) and chapter 15.II, one can deduce that Cbl(II) binds O2 with the following generation of reactive oxygen species (HO2- and H2O2), which in turn oxidize Cbl(II), giving a considerable fraction of inactive yellow corrinoids. If the authors found any useful information under their investigation of the literature, they are very welcome to share it with the readers.

Answer #6. We agree with Reviewer #1. We have now elaborated in the Discussion section the mechanistic aspects related to N2O-induced toxicity in the setting of one-carbon metabolism alterations, as follows: “Several mechanistic hypotheses have been advanced for explaining the association between N2O exposure and the occurrence of clinical and biochemical abnormalities associated with vitamin B12 deficiency, notably subacute combined degeneration of the spinal cord in association with hyperhomocysteinemia and elevated levels of methylmalonic acid. Hathout & El-Saden nicely described these hypotheses in their review paper regarding N2O-induced myelopathy [23]. Three main mechanistic hypotheses have been put forward as the knowledge in the field progressed: the alteration of the MMCoAM pathway[23], the alteration of the methylcobalamin-MTR pathway[23,27,28], and more recently, the imbalance between cytokines and growth factors exhibiting myelinotoxic or myelinotrophic effects [29-33]. Early observations initially suggested that the toxicity of N2O occurred primarily through the MMCoAM pathway. However, this was challenged by clinical observations in patients with monogenic inherited disorders associated with methylmalonic acidemia who do not develop subacute combined degeneration [23]. The second hypothesis underlying the relationship between N2O exposure and neurological complications is the effect of N2O on the methylcobalamin-MTR pathway. Indeed, inherited disorders in MTR activity induce a decrease of methionine and a low methyl-donor status with defective methylation and resulting instability of the myelin sheath. However, this hypothesis was hampered by experiments using the B12-deficient fruit bat animal model. These experiments did not show significant alterations in S-adenosyl-methionine and S-adenosyl-homocysteine in the brain and spinal cord after induction of severe vitamin B12 deficient myelopathy following a combination of dietary deprivation and N2O exposure [23,27,28]. During the last two decades, much progress has been made in the field of neuroimmunology and have suggested that the pathogenesis of N2O-induced myelopathy could be explained by an imbalance between cytokines and growth factors exhibiting myelinotoxic (tumor necrosis factor alpha, sCD40:sCD40L dyad, nerve growth factor) or myelinotrophic (interleukin-6 and epidermal growth factor) effects [29-33].

***

Comment #7. Line 38. Change the text to “… and propellant for the preparation of whipping cream.”

Answer #7. We thank Reviewer #1 for the comment, and we apologize for the typo error. We have updated the manuscript accordingly.

***

Comment #8. Line 386. Change the text to “… were not available for all patients …”

Answer #8. We thank Reviewer #1 for the comment, and we apologize for the typo error. We have updated the manuscript accordingly.

***
